# The Impact of Vaccination on COVID-19 Burden of Disease in the Adult and Elderly Population: A Systematic Review of Italian Evidence

**DOI:** 10.3390/vaccines11051011

**Published:** 2023-05-22

**Authors:** Giovanna Elisa Calabrò, Ciro Pappalardo, Floriana D’Ambrosio, Michele Vece, Chiara Lupi, Alberto Lontano, Mattia Di Russo, Roberto Ricciardi, Chiara de Waure

**Affiliations:** 1Section of Hygiene, Department of Life Sciences and Public Health, Università Cattolica del Sacro Cuore, 00168 Rome, Italy; giovannaelisa.calabro@unicatt.it (G.E.C.); floriana.dambrosio01@icatt.it (F.D.); alberto.lontano01@icatt.it (A.L.); mattia.dirusso88@gmail.com (M.D.R.); 2VIHTALI (Value in Health Technology and Academy for Leadership & Innovation), Spin-Off of Università Cattolica del Sacro Cuore, 00168 Rome, Italy; robertoricciardi.mail@gmail.com; 3Department of Medicine and Surgery, University of Perugia, 06132 Perugia, Italy; michele.vece@studenti.unipg.it (M.V.); chiara.lupi1@studenti.unipg.it (C.L.); chiara.dewaure@unipg.it (C.d.W.)

**Keywords:** COVID-19, SARS-CoV-2, vaccination, vaccines, mortality, lethality, hospitalization, complications, adult population, elderly people

## Abstract

COVID-19 is a major global health threat, with millions of confirmed cases and deaths worldwide. Containment and mitigation strategies, including vaccination, have been implemented to reduce transmission and protect the population. We conducted two systematic reviews to collect nonrandomized studies investigating the effects of vaccination on COVID-19-related complications and deaths in the Italian population. We considered studies conducted in Italian settings and written in English that contained data on the effects of vaccination on COVID-19-related mortality and complications. We excluded studies that pertained to the pediatric population. In total, we included 10 unique studies in our two systematic reviews. The results showed that fully vaccinated individuals had a lower risk of death, severe symptoms, and hospitalization compared to unvaccinated individuals. The review also looked at the impact of vaccination on post-COVID-19 syndrome, the effectiveness of booster doses in older individuals, and nationwide adverse events. Our work highlights the crucial role that vaccination campaigns have played in reducing the burden of COVID-19 disease in the Italian adult population, positively impacting the pandemic trajectory in Italy.

## 1. Introduction

COVID-19, caused by the SARS-CoV-2 virus, has emerged as a significant global health threat [1]. Since its initial identification in Wuhan, China, in December 2019, COVID-19 has spread worldwide, resulting in a devastating impact on public health and the economy [1]. As of 29 March 2023, there have been over 761 million confirmed cases and more than 6.8 million deaths globally, with Europe alone accounting for over 2.2 million deaths [2]. The clinical outcomes of COVID-19 vary among individuals and are influenced by factors such as age, gender, ethnicity, and underlying health conditions [3]. Symptom presentation ranges from asymptomatic cases to severe complications posing a risk to life [3]. Older adults, individuals with pre-existing health conditions, and those with weakened immune systems are particularly vulnerable to severe infection and associated mortality [3,4]. Complications can arise during SARS-CoV-2 infection, including pneumonia, respiratory failure, acute respiratory distress syndrome (ARDS), systemic inflammatory response syndrome (SIRS), uncontrolled inflammatory response, hypercoagulation, immune system dysfunction, acute liver injury, hypoproteinemia, and organ failure. These complications can significantly impact the severity of the disease and contribute to COVID-19-related hospitalizations and deaths [5]. A study conducted by the Italian National Health Institute (Istituto Superiore di Sanità, ISS) analyzed 52,556 Italian death certificates from March and April 2020 and found that the majority of complications leading to death were related to the respiratory system. Specifically, pneumonia and acute respiratory failure were the most common respiratory complications, while ARDS was less frequently reported. Among nonrespiratory complications, sepsis and renal failure were the leading causes of death, with pulmonary embolism being among the other contributing factors [6]. In response to the pandemic, numerous containment and mitigation interventions have been put in place to prevent overwhelming healthcare systems and protect vulnerable populations [7]. Measures, such as social distancing, hand hygiene, masks, quarantines, and lockdowns, have proven effective in reducing SARS-CoV-2 transmission, but vaccination has also played a crucial role in minimizing the risk of serious illnesses from COVID-19 [3,8,9,10,11].

The published data suggest that achieving high vaccine coverage is the primary factor in reducing the incidence of SARS-CoV-2 infections, both in terms of clinical disease and more severe outcomes, such as hospitalization and death [12,13]. Continued monitoring and long-term surveillance are important to understand the effectiveness of SARS-CoV-2 vaccination against symptomatic and serious infections [14,15].

Through an extraordinary global scientific endeavor, several vaccines have been developed to combat the escalating impact of COVID-19 [16]. To date, the European Medicines Agency (EMA) and the Italian Medicines Agency (AIFA) have approved six COVID-19 vaccines (Nuvaxovid, Vidprevtyn Beta, Jcovden, Vaxzervria, Spikevax, Comirnaty), with ongoing investigations exploring various vaccine platforms, including inactivated vaccines, live attenuated vaccines, viral vector-based vaccines, subunit (recombinant protein) vaccines, DNA-based vaccines, and RNA-based vaccines [3,17,18,19]. Italy, as the first Western country affected by the COVID-19 pandemic, has reported over 25.6 million cases and 189,000 deaths as of 29 March 2023 [2,20]. Since the detection of the initial cases in February 2020, the pandemic has exerted significant strain on national and regional healthcare systems, imposing a significant burden in terms of morbidity and mortality, especially among the elderly population [21]. The consequences on health systems, on the economy of different countries, and on society have been profound [22].

Italy began its COVID-19 vaccination campaign in December 2020 with the Pfizer-BioNTech Comirnaty vaccine, targeting healthcare professionals and long-term care residents [23]. Since then, the country has approved the use of other vaccines [24]. The national vaccination protocol has undergone several changes since its inception. In January 2021, Italy expanded the vaccine rollout to include additional priority groups, such as teachers, police officers, and individuals with underlying health conditions. In May 2021, Italy approved the use of the Pfizer-BioNTech vaccine in individuals aged 12 to 15 years [25]. Shortly after, in June 2021, the “green pass” system was implemented [26]. This system mandated that the general public present evidence of vaccination or a negative COVID-19 test result to attend specific events. However, the use of the green pass was discontinued in May 2022 following the conclusion of the COVID-19 state of emergency [27]. In September 2021, booster doses were officially approved and initially offered to older age groups. Later on, booster shots were made available to individuals aged 60 and over, healthcare workers, and those with underlying health conditions [28]. Primary vaccination is still currently recommended for all individuals aged five and above, and booster shots are recommended for various groups, including individuals aged 12 and above who have not received a booster shot yet, healthcare workers, elderly individuals, and those with underlying medical conditions [25,29]. As of 29 March 2023, a total of 144,341,632 vaccine doses have been administered [30,31]. The most frequently administered vaccines were Comirnaty and Spikevax, followed by Vaxzervria and Jcovden. Notably, Comirnaty was the most commonly administered vaccine by a significant margin [31]. During the vaccination campaign, a multitude of data from randomized controlled trials (RCTs) and real-world observational studies emerged, providing insights into the efficacy and safety of registered vaccines globally [32]. The evidence available today demonstrates the effectiveness of COVID-19 vaccines in reducing the risk of developing symptomatic disease, surpassing the efficacy rates of other vaccines, such as influenza vaccines (around 50–60% efficacy), and effectively preventing severe disease, hospitalizations, and deaths [22,33,34] associated with COVID-19. In 2021, across 90 countries, Liang et al. [35] showed that for every 10% increase in COVID-19 vaccine coverage, the mortality rate could be reduced by 8%. In Italy, the last Technical Note issued by the ISS estimated that just over 8 million infections, over 500,000 hospitalizations in medical units, over 55,000 hospitalizations in intensive care units (ICUs), and over 150,000 deaths were directly avoided by COVID-19 vaccination in the period January 2021–January 2022 [36]. Ongoing monitoring of the effectiveness and safety of COVID-19 vaccination is necessary, as it may be influenced by features such as waning immunity and the emergence of more transmissible variants [37]. Moreover, evaluating the effectiveness of COVID-19 vaccines in preventing severe disease is critical, as this outcome imposes the greatest burden on healthcare systems and drives most policy decisions related to COVID-19. Such evaluation is essential to supplement the evidence from RCTs and to provide guidance for public health recommendations [38,39]. This review is aimed at providing an overview of nonrandomized evidence on the impact of COVID-19 vaccination on complications resulting from SARS-CoV-2 infection and related deaths among adults and the elderly based on the Italian literature.

## 2. Materials and Methods

### 2.1. Search Strategy

Two systematic reviews were conducted to evaluate the impact of vaccination on the COVID-19 burden, specifically in terms of related complications and deaths. These reviews were conducted in accordance with the Preferred Reporting Items for Systematic Reviews and Meta-Analyses (PRISMA) guidelines [40]. Our systematic reviews were not registered in any review registry, and we did not prepare a review protocol for this study.

PubMed was the only database utilized as the primary source for the retrieval of studies. A comprehensive search strategy was employed, utilizing specific keywords and filters, on 22 September 2022. The search strings were developed and executed using PubMed’s preset filters (as detailed in Table 1).

### 2.2. Inclusion and Exclusion Criteria

All nonrandomized studies containing data on the effects of vaccination on COVID-19-related mortality and complications were considered eligible. The inclusion criteria for the studies were based on the location, language, and population age. Studies that were conducted in non-Italian settings and that pertained to the pediatric population were excluded, as well as articles that could not be located or lacked sufficient information for the purposes of the research as well as articles that were not written in English.

### 2.3. Selection Process and Data Extraction

Records were entered into a dedicated Excel spreadsheet and/or processed in Rayyan (a web-based study selection tool) [41], and, after duplicates were removed, they were evaluated using inclusion/exclusion criteria. Articles were initially screened by title and abstract, and then full texts of the papers deemed eligible were read. The articles were also subjected to a snowballing process, which involved examining the bibliographic references and citations to find additional papers that met the inclusion criteria. Risk of bias assessment was not conducted in this study, as no standardized tool was used. However, to ensure the reliability of the included studies, six reviewers independently reviewed the articles found in the literature search (F.D.A., M.D.R., A.L., C.L., C.P., and M.V.), and disagreements were resolved by two senior researchers (G.E.C. and C.d.W.).

The included studies were then entered into summary tables as part of the data extraction process. For each review, a separate table was created to better systematize the main findings. The tables included information, such as the first author’s name, year of publication, study setting, study design, sample characteristics, study period, study aim, results, and main findings, which were then summarized qualitatively.

#### Data Synthesis

The synthesis of studies included in the two systematic reviews was based on a categorization scheme of their characteristics and outcomes.

Specifically, studies included in the section on the impact of vaccination on COVID-19 mortality were grouped based on their examination of COVID-19 mortality and the effects of vaccination on this outcome in diverse populations. In particular, these studies focused on mortality rates in vaccinated and unvaccinated populations, with a special emphasis on subgroups, such as the elderly and those with underlying health conditions.

Similarly, studies included in the section on the effects of vaccination on COVID-19 complications were synthesized based on their focus on COVID-19 severity and hospitalizations. For COVID-19 severity, patients were stratified based on the severity of their symptoms, which was classified as absent/minimal, mild, or severe/critical. The studies then reported the percentage of patients in each category based on their vaccination status, which was classified as fully vaccinated (FV), partially vaccinated (PV), or unvaccinated (UV). In the case of hospitalizations, the studies reported the percentage of patients that were treated in an inpatient, outpatient, or ICU setting based on their vaccination status.

## 3. Results

In order to increase comprehension of the results, this section reports the results of the two systematic reviews separately. The study periods of all included articles are shown in Figure 1.

### 3.1. Effects of Vaccination on COVID-19-Related Mortality

Table 2 provides a comprehensive summary of the variables collected from the papers included in this systematic review and is, therefore, referenced in this section for a clearer understanding of the data.

#### 3.1.1. Characteristics of Studies Included in the Systematic Review on COVID-19 Mortality

As shown in Figure 2, 1529 records were screened, of which 1504 were excluded after title and abstract screening, and 25 were assessed for eligibility. Subsequently, 17 studies were excluded because they were not related to the topic, and a further two studies did not have data on the Italian population. Only one study was retrieved through the snowballing research process [42]. A total of seven studies [42,43,44,45,46,47,48] were included: five of them were published in 2022 [43,44,45,47,48], while the rest were published in 2021 [42,46]. The research was conducted in different locations: one was a multinational study (40 countries, including Italy) [47], two analyzed data from the Italian peninsula [43,46], one included the residents of Milan and Lodi [45], and three evaluated nursing home (NH) residents of the Veneto region [44] and Florence [42,48]. The study sample of the articles ranged from 2271 [48] to 4,487,526 [43] subjects, with a mean of 1,499,612 and a median of 852,211 subjects. A variety of aims was pursued, with two addressing the effectiveness of the vaccine booster dose in reducing the impact of SARS-CoV-2 in terms of hospital admissions, ICU admissions, and deaths [43,45]; three articles describing the impact of vaccination on NH residents [42,44,48]; one assessing the impact of vaccination in terms of averted COVID-19 cases, hospitalizations [46], ICU admissions, and deaths by age group and geographical macroareas [46]; and another analyzing the proportion of deaths in different age groups in vaccination versus control periods [47]. Considering 27 December 2020 as the start date of the vaccination campaign in Italy, three studies compared pre- and postvaccination periods [42,44,47], while others only used data from the postvaccination period (Figure 1). The observation periods ranged from a minimum of 91 [45] to a maximum of 465 days [42,47], with a mean of 283.4 and a median of 262 days.

#### 3.1.2. Results of Studies Included in the Systematic Review on COVID-19 Mortality

Rivasi et al. investigated the impact of COVID-19 vaccination on NH residents of the Florence Health District network. The 2021 study [42] included 3730 people and analyzed the period starting from 1 October 2020 and ending on 31 March 2021. Fully vaccinated subjects (those who have received the required doses of any of the approved vaccines, FV) showed significantly lower mortality rates than partially vaccinated (those who received only one dose of the approved vaccines, PV) and unvaccinated subjects (UV) (mortality: FV 6%, PV 18%, UV 56%). The risk of death was 84% and 96% lower in PV (HR 0.157, 95% CI 0.049–0.491) and FV subjects (HR 0.037, 95% CI 0.006–0.223) versus UV subjects. The second study [48] was based on a sample of 2271 vaccinated residents and investigated the period between 1 April 2021 and 31 October 2021 with the aim of describing the long-term effect of vaccination. The mortality rate within six months after the primary vaccination course was 5%, which increased to 8% during the following three months, although the mortality rate among vaccinated individuals was lower than the previously reported rates in the unvaccinated residents of the study area (31.1% in March–April 2020 and 23% in October 2020–January 2021).

Pierobon et al. [44] examined the risk of COVID-19 hospitalization and death among NH residents in comparison to the general population in the same age bracket in the Veneto region. The study period spanned from 21 February 2020 to 3 May 2021, divided into three time windows corresponding to the COVID-19 waves (the first wave from 21 February 2020 to 31 August 2020; the second wave from 1 September 2020 to 1 January 2021; and the third wave from 1 February 2021 to 3 May 2021). The study included 852,211 residents aged between 70 and 100 years old, of which 33,592 were NH residents, and the sample was matched based on propensity scores. Ultimately, the study analyzed 31,922 subjects.

Considering the overall period, NH residents showed a higher risk of death (relative risk (RR) = 6.07), but by the end of the vaccination program (NH residents and healthcare workers were among the first ones to be vaccinated in Italy), the RR was significantly reduced. The RRs were 10.10 in the first wave, 7.97 in the second wave, and 0.48 in the third wave.

An article published by Mattiuzzi and colleagues in 2022 [43] investigated the effectiveness of vaccination booster doses in Italian people aged ≥80 years (4,487,526), using statistics retrieved from the official bulletin published on 17 December 2021 by the ISS. This institutional document presents comprehensive nationwide epidemiological data on the impact of COVID-19, including the number of cases, hospitalizations, ICU admissions, and deaths. It also provides information on the progress of the vaccination campaign against COVID-19 in Italy. According to the results, individuals who received an additional shot had a 98% and 81% lower risk of death compared to the unvaccinated ones and to those who only completed the primary COVID-19 vaccination cycle, respectively.

Similarly, Russo et al. [45] studied the efficacy of the booster dose on 2,936,193 residents of the Milan and Lodi provinces aged 19 years or over in the period from 1 October 2021 to 31 December 2021. The cohort was stratified into ten levels based on the combination of SARS-CoV-2 infection, the number of shots received, and the time elapsed since the last vaccination. In comparison with the group of people that received an additional vaccine dose, unvaccinated subjects had a three-fold risk of dying.

Sacco et al. [46] conducted a study to evaluate the impact of COVID-19 vaccination on the prevention of infections, hospitalizations, ICU admissions, and deaths. The study covered the period from 11 January 2021 to 30 September 2021 and analyzed different age groups (<60, 60–69, 70–79, and ≥80 years old) and geographical macroareas. The study estimated that the vaccination campaign averted a total of 22,067 deaths. The majority of the prevented deaths (71%) were in individuals aged 80 years and older, while 18%, 8%, and 2% were averted in those aged 70–79, 60–69, and under 60 years old, respectively. Vaccine uptake varied across different regions and macroareas in Italy (north-west, north-east, center, and the south islands). Geographical areas with higher and faster vaccination rates were able to prevent a greater number of deaths during the summer months. Without vaccination, the expected mortality rate for individuals aged 80 years and older would have been 224 per 100,000 compared to the observed rate of 32 per 100,000 during the same period. It was especially notable in the South and the islands with a lower difference (157 vs. 52) and in the Centre with a larger difference (332 vs. 27). The south and the islands experienced fewer averted events due to a slower vaccination uptake among high-risk individuals and the high incidence of COVID-19 cases during the summer season (July–August).

A multinational study conducted by a group of researchers from Università Cattolica del Sacro Cuore and Stanford University [47] in 2022 performed a meta-analysis to assess the proportion of deaths across different age groups (0–49, 50–69, and ≥70 years old) during the vaccination period (considered to be between 14 January 2021 and 26 May 2021) compared to two control periods (the entire prevaccination period and the second COVID-19 wave period starting on 30 August 2020). The study found that countries, such as Italy, that prioritized vaccination among older individuals experienced a significant shift in the age distribution of COVID-19 deaths during the first five months of 2021. When comparing the proportion of COVID-19 deaths in the 0–69 age group to the general population, this group represented 14% of the total in the entire prevaccination period, 13% in the second epidemiological wave, and 16% during the vaccination period. Compared to both the entire prevaccination period and the prevaccination period of the second wave, there was a noticeable increase in the proportion of deaths among the nonelderly population (0–69 years old) during the vaccination period. The proportion ratios (PR) were 1.15 (95% CI, 1.12–1.18) and 1.20 (95% CI, 1.15–1.24), respectively.

### 3.2. Effects of Vaccination on COVID-19-Related Complications

Table 3 provides a comprehensive summary of the variables collected from the papers included in this systematic review and is, therefore, referenced in this section for a clearer understanding of the data.

#### 3.2.1. Characteristics of Studies Included in the Systematic Review on COVID-19 Complications

For this review, 3694 records were assessed for eligibility: 3678 were excluded after screening by the title and abstract, and another eight were excluded because they were not related to the topic under study or because the study type did not meet the inclusion criteria. Eventually, nine studies investigating the effect of vaccination on COVID-19-related complications were included (Figure 3). Out of the nine records, eight were found through the review of the available literature [43,44,45,46,48,49,50,51], and one was retrieved through the snowballing research process [42]. Six out of the nine articles were in common with the systematic review on mortality [42,43,44,45,46,48].

Seven articles (78% of the total) [42,44,45,48,49,50,51] were published in 2022, with only two (22%) published in 2021 [42,46]. Four (44%) [44,48,49,50] were prospective studies, while the remaining five (56%) [42,43,45,46,51] were retrospective studies. The study sample ranged from 87 [49] to 4 487 526 subjects [46], with locations primarily in northern Italy (Florence [42,48], Udine [50], the Milan and Lodi provinces [45], and the Veneto region [44,51]) and central-southern Italy (Campobasso [49]); two studies (22%) were based on nationwide data [43,46], while four articles (44%) investigated NH residencies [42,43,48,49]. The studies investigated data from the years 2020 (*n* = 2; 22%) [42,44] and 2021 (*n* = 7; 78%) [43,45,46,48,49,50,51] (Figure 1). Four articles (44%) described the effect of vaccination on NH residents in terms of the risk of hospitalization and COVID severity [42,44,48,49], one (11%) was a descriptive analysis of ICU admissions for severe COVID-19 after the local vaccination campaign [51], one (11,1%) followed the impact of vaccination on post-COVID-19 syndrome [50], two (22%) provided an analysis on the effectiveness of COVID-19 vaccine booster doses in preventing severe forms of diseases [43,45], and one (11%) estimated the nationwide averted events (infections, hospitalizations, and deaths) [46].

#### 3.2.2. Results of Studies Included in the Systematic Review on COVID-19 Complications

Out of the nine included articles, three stratified COVID-19 infection cases by severity [42,48,50]. Based on the symptoms, patients were subsequently divided into three groups: absent/minimal, mild (fever, cough, flu-like symptoms, diarrhea, no pneumonia), or severe/critical (respiratory failure requiring oxygen therapy, severe gastrointestinal symptoms, including severe anorexia, delirium). One of the studies conducted on NH residents in Florence Health District reported that severity was absent/minimal in 22% of the unvaccinated (UV) subjects, mild in 33%, and severe in 44% of cases. On the other hand, 45% of the partially vaccinated (PV, who received at least one shot) and 86% of the fully vaccinated (FV, who completed the primary vaccination course) groups experienced absent/minimal symptoms, while 39% and 14%, respectively, had a mild form of the disease. During the study, 16% of the population that had received the first shot and 0% of the group that completed at least the primary vaccine cycle had severe symptoms [42]. Another study by the same authors showed that among the FV population, severity was classified as absent/minimal in 58% of the cases, mild in 25%, and severe in just 17% [48]. Lastly, an observational prospective study conducted in Udine found that 6% of the unvaccinated group was asymptomatic, 69% had mild symptoms, and 26% were in severe/critical conditions. For the vaccinated group who received a full primary vaccination cycle, 14%, 65%, and 21% were, respectively, asymptomatic, mildly symptomatic, and severely or critically symptomatic [50] (Table 4).

#### 3.2.3. Hospitalizations for COVID-19

Data regarding hospitalizations caused by COVID-19 were reported by eight studies [42,43,44,45,46,48,49,50]. Peghin et al. [50] conducted a study on 479 patients treated for COVID-19 and reported that in both vaccinated (primary cycle concluded) and unvaccinated populations, the majority were treated in an outpatient setting (75% vs. 69%, respectively), while 2% of the people that were vaccinated and 5% of the unvaccinated group were admitted to ICUs; the remainder of the cases were treated in an inpatient setting (4% of the vaccinated and 5% of the unvaccinated). A study conducted on 3730 SARS-CoV-2-naïve NH residents (mean age 84.2 years) by Rivasi et al. [42] showed that hospitalization was needed in 33% of the UV population under study, in 14% of the PV population, and in 3% of the FV population; no data were reported on whether ICU admissions were needed or not. Another study [48] conducted on 2271 vaccinated (primary cycle) NH residents (mean age 86.6 years) by the same authors reported a total of eight cases of hospitalizations among FV residents (no data were mentioned about ICU admissions), corresponding to 8% of the infected subjects and 0.35% of the population under investigation. In the six months following the primary immunization course (1 April 2021–31 July 2021), the hospital admission rate was 10%; thereafter (1 August 2021–31 October 2021), it was 7%. A prospective study from Ripabelli et al. [49] mentioned three hospitalizations (3% of the study sample, 8% of the infected ones, 4% of the NH residents, 0% of the workers) and zero ICU admissions among a group of 87 FV (primary cycle completed or additional dose) NH residents and workers. All the three hospitalized patients were NH residents. An article using data on the Italian population of 80 years old or older reported that in this cohort, the completion of the full primary vaccination cycle was linked with an 86% reduced risk of COVID-19 hospitalization and ICU admission compared to nonvaccination within five months. The effectiveness against the risk of COVID-19-related hospitalization and ICU admission was 83% and 86%, respectively, after five months, but this protection tended to fade after that point. Those who received the COVID-19 vaccine booster dose had an 83% lower risk of COVID-19 hospitalization and an 82% lower risk of ICU admission compared to those who had completed the full primary vaccination cycle for more than five months. When compared to individuals who had recently (i.e., within the past five months) completed the full primary vaccination cycle, the booster dose produced higher protection against all these outcomes (a 79% lower risk of hospitalization and an 87% lower risk of ICU admission) [43]. Another study conducted in Veneto by Pierobon et al. [44] described the impact of the vaccination campaign on the population living in the local NH (age range 70–100 years old) in comparison with the general population within the same age group after 1:1 propensity score matching. As said earlier, the study period (21 February 2020–3 May 2021) was characterized by the identification of three epidemiological waves (the first wave up to 31 August 2020; the second wave up to 1 February 2021; and the third wave up to 3 May 2021), and the third wave was utilized to evaluate the efficacy of the vaccination campaign. The following results were reported: the RRs of hospitalization for COVID-19 were 4.89 (95% CI: 4.17–5.74) for the first wave, and 2.50 (95% CI: 2.29–2.73), and 0.25 (95% CI: 0.16–0.38) for the second and third wave, respectively. In a study by Russo et al. [45], the effectiveness of the booster dose was evaluated among 2,936,193 residents of the Milan and Lodi provinces who were 19 years or older between 1 October 2021 and 31 December 2021. The study cohort was divided into ten groups based on SARS-CoV-2 infection status, the number of vaccinations received, and the time since their last vaccination, as previously discussed. The results showed that individuals who did not receive the booster dose had a 10-fold higher risk of hospitalization and a 9-fold higher risk of ICU admission compared to those who received the additional dose. Ultimately, according to the previously cited retrospective study from Sacco et al. [46] conducted on the entire Italian population over 60 years old, the vaccination campaign was estimated to have prevented 60,659 hospitalizations and 8056 ICU admissions, accounting for 36% and 32% of the expected events (observed plus averted), respectively. Most of the avoided events involved people ≥70 years old (Table 5; the COVID-19 vaccination status among adults admitted to ICU in Veneto was discussed on JAMA Network Open [51]). This retrospective article collected data from several hospitals and healthcare institutions pertaining to the Veneto ICU Network. A total of 748 patients aged 18 years or older (mean age of 62 years) admitted to the ICU from May to December 2021 for COVID-19–associated acute respiratory distress syndrome were included in the study. A total of 552 people (74%) were not vaccinated, 138 (18%) were partially vaccinated (one dose), and 58 (8%) completed at least the primary vaccination cycle. For PV patients, the median time from vaccine administration to ICU admission was 22.5 days (interquartile range—IQR, 16.0–49.8 days), while for FV patients, that period was 159.0 days (IQR, 112.0–192.0 days). The same was confirmed considering the median time to hospital admission, with a median of 154.0 days (IQR, 110.0–190.0 days) in those vaccinated that completed the primary vaccination cycle and 16.0 days (IQR, 11.0–32.5 days) in the group that did not complete the vaccine course. The trend of ICU admissions per million of the Veneto region residents on 1 January 2021 in FV, PV, and UV patients over the calendar study months was evaluated using the Cox–Stuart method. The distribution of the events according to the vaccination status was estimated using a generalized linear model and the Wald chi-squared test. ICU admissions among UV patients showed a statistically significant rising trend. In contrast, for those who had received vaccinations, the tendency stayed the same.

#### 3.2.4. Post-COVID-19 Syndrome

Peghin et al. [50] described the impact of vaccination on post-COVID-19 syndrome. Interviews were conducted at 6 and 12 months after disease onset in 479 adults, representing the whole population of the in- and outpatients with COVID-19 at Udine Hospital in the March–May 2020 period. In total, 132 out of 479 completed the primary vaccination cycle, and the median length of in-hospital stays was similar for both groups (6.5 days in the vaccinated subjects and 7 days in the nonvaccinated). At six months, 44 (33%) vaccinated people had experienced it, while the number was higher among unvaccinated people (157, 45% of the total); at 12 months, the majority of participants reported that they were still unaffected or their health status had not changed compared to the previous follow-up (66% and 71%, respectively). However 23% of the vaccinated group and 16% of the nonvaccinated group indicated a worsening of their clinical condition, and 11% and 13% reported an improvement.

## 4. Discussion

In our systematic review, the data from 10 studies showed that the COVID-19 vaccination of the adult and elderly Italian population was associated with three main findings:A reduction in the risk for COVID-19-related death in subjects who were fully vaccinated and/or had received a booster when compared to unvaccinated or partially vaccinated subjects;A reduction in hospitalizations and ICU admissions for COVID-19 in subjects who were fully vaccinated and/or had received a booster dose when compared to unvaccinated or partially vaccinated subjects;A reduction in the clinical severity of COVID-19 in subjects who were fully vaccinated and/or had received a booster dose when compared to unvaccinated or partially vaccinated subjects.

Overall, these results confirm the effectiveness of COVID-19 vaccines in preventing critical outcomes related to the disease. The approved COVID-19 vaccines have been shown to elicit similar levels of neutralizing antibodies and robust memory B cell responses, which play a critical role in preventing severe illness and hospitalization. Wang et al. found no significant differences in the ability of Pfizer-BioNTech, Moderna, and Johnson & Johnson vaccines to induce neutralizing antibodies against the original SARS-CoV-2 virus or its variants [52]. Saadat et al. also reported comparable neutralizing antibody responses induced by the Pfizer-BioNTech and Moderna vaccines [53]. Moreover, Goel et al. demonstrated that the Pfizer-BioNTech and Moderna vaccines induced robust and persistent memory B cell responses for at least 6 months after the second dose [54]. These findings suggest that the different immune arms elicited by COVID-19 vaccines play a critical role in their protective effectiveness against the virus.

The majority of the research articles included in this review focused on the older population [42,43,44,48,49], who are among the most vulnerable groups and were the first to be vaccinated in Italy. In accordance with the current literature [55,56,57,58], we found that most of the averted critical outcomes were related to those 80 years old or older [46], and there was a shift in the age distribution of COVID-19 deaths in the first five months of 2021, which can be attributed to the prioritization of vaccination among older individuals [47].

Our findings showed that the risk of death among vaccinated subjects was consistently reduced in comparison to unvaccinated individuals [45] and often reduced to almost 100% in the elderly population [42,43]. This is in agreement with the findings of other studies conducted internationally in both the fully vaccinated and those who received a booster dose [14,59,60,61]. Regarding the duration of the protection provided by the vaccine, Russo et al. reported that the risk of death due to COVID-19 was significantly reduced when two doses in the previous five months or a booster were received [45]. While this is partially supported by the current literature, uncertainty remains regarding the waning of vaccine effectiveness and the impact of virus variants [39,60,62,63,64,65].

Similar to the findings on death, the risk of hospitalization and ICU admission was found to be lower in people who were vaccinated in comparison to those who were unvaccinated [43]. This trend was consistently observed across multiple studies, with hospitalizations and ICU admissions consistently lower in fully vaccinated individuals compared to unvaccinated individuals [42,48,49,50]. Additionally, the administration of a booster dose was shown to significantly reduce the risk of hospitalization and ICU admission for COVID-19, particularly when administered six or more months after the completion of the primary vaccination cycle [45]. Our findings are consistent with those from the studies conducted in European and non-European settings [14,59,64,66,67,68].

The effectiveness of vaccination on the clinical severity of COVID-19 was examined in a limited number of studies [42,48,50]. Nevertheless, the current evidence is generally poor and often based on different criteria or definitions of severity. One study conducted in the United States using the World Health Organization’s (WHO) clinical progression scale found that vaccination was associated with a reduced risk of progression to invasive mechanical ventilation [65]. Another study based on a modified version of the WHO scale found that mRNA vaccines may attenuate disease severity among patients who develop COVID-19 despite vaccination [69]. A study conducted in Singapore evaluated the disease as severe (i.e., requiring oxygen supplementation, ICU admission, or leading to death) or not severe and concluded that the primary vaccination cycle and booster dose were associated with an increase in protection against severe COVID-19 [70]. In line with these findings, the results of our review showed that COVID-19 clinical severity was less in fully vaccinated individuals and tended to be mild or severe in partially vaccinated and unvaccinated individuals [42,48,50].

It is important to note that a difference in effectiveness according to the number of received doses arose from the included studies [43,45]. This may be attributed to the fact that a two-dose regimen may be necessary for some vaccines to achieve an optimal level of protection. According to the current evidence, administering a single dose of the vaccine can result in a partial immune response and a more limited period of protection compared to receiving all recommended doses [67,71]. Furthermore, the need for booster doses to maintain immunity over time has also been demonstrated in several studies. For instance, a study on the Pfizer-BioNTech vaccine found that while two doses were highly effective in preventing COVID-19, vaccine efficacy decreased over time, particularly against the Delta variant, and a booster dose significantly improved protection [62]. Another study on the Moderna vaccine similarly showed that a booster dose significantly increased antibody levels and improved protection against variants of concern [72].

The effects of vaccinations on post-COVID-19 syndrome were investigated in one study with inconclusive results [50]. Nevertheless, the current evidence suggests that vaccination might be protective against post-COVID-19 and long COVID-19 syndromes [73,74], even though there is a lack of standardized definitions and short follow-up periods.

It is important to acknowledge that the results of this systematic review are subject to certain limitations. First and foremost, all the included studies are observational and not RCTs, which may affect the robustness of the findings. Additionally, the studies included in this review exhibit a degree of heterogeneity in terms of their designs, settings, and outcomes, which may affect the comparability of the results. Furthermore, the study period and follow-up times varied across the included studies, which may have resulted in different SARS-CoV-2 variants circulating in Italy at the time of the study and in different phases of the vaccination campaign. Furthermore, our systematic review identified only one published study on post-COVID-19 syndrome, which may limit the extent to which our findings on this topic can be applied to other populations or settings. Additionally, because there was no differentiation among the administered vaccines, the applicability of our results may be further constrained. The systematic reviews consulted only one database; therefore, other available studies could have been missed, and a potential selection bias could not be completely ruled out.

Nevertheless, to the best of our knowledge, this is the first systematic review that provides a systematization of real-world evidence on the effectiveness of COVID-19 vaccination in respect to several critical outcomes in the Italian population. The synthesis of this kind of evidence complements the available information obtained from RCTs.

## 5. Conclusions

The findings from this systematic review demonstrate the crucial role that vaccination has played in reducing the burden of COVID-19 disease in the Italian adult and elderly population. The review of the included studies highlighted the protective effect of vaccination against COVID-related complications and deaths, particularly among those who have received the full primary vaccination and booster doses. The implementation of a mass vaccination campaign has significantly reduced the strain on the healthcare system and society, thereby positively impacting the pandemic trajectory in Italy.

It is evident that vaccination should continue to be prioritized to prevent severe forms of the disease in future epidemic peaks, especially among high-risk individuals. However, given the ongoing evolution of the pandemic, the emergence of new variants of the virus, and the ongoing research on new drugs, the continuous generation and collection of evidence are crucial in ensuring that the population remains protected. Indeed, the strengthening of the evidence and the generating of new data will be essential to support evidence-based vaccination policies that recognize the full value of vaccines and vaccination [75]. Therefore, it is crucial that public health policymakers remain vigilant in monitoring data on COVID-19 and adapt interventions accordingly through the active and informed involvement of all relevant stakeholders, including citizens.

## Figures and Tables

**Figure 1 vaccines-11-01011-f001:**
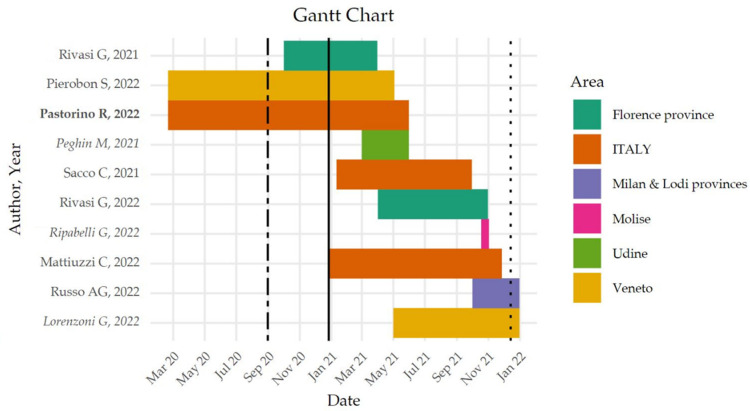
Included studies’ periods and areas. This figure shows the timeline and area of included studies from March 2020 to January 2022 [42,43,44,45,46,47,48,49,50,51], with the beginning of the vaccination campaign marked by a solid line, the one on the left dividing the first and second waves, and the transition from the Alpha/Delta variants to the Omicron variant marked by a line on the right. Overlapping studies were present in the two systematic reviews, resulting in a total of 10 unique studies. Among the total of 10 unique studies, the author names and publication years are in bold for studies on the impact of vaccination on COVID-19 deaths exclusively included in the review and italicized for studies on the impact of vaccinations on COVID-19-related complications exclusively included in the review.

**Figure 2 vaccines-11-01011-f002:**
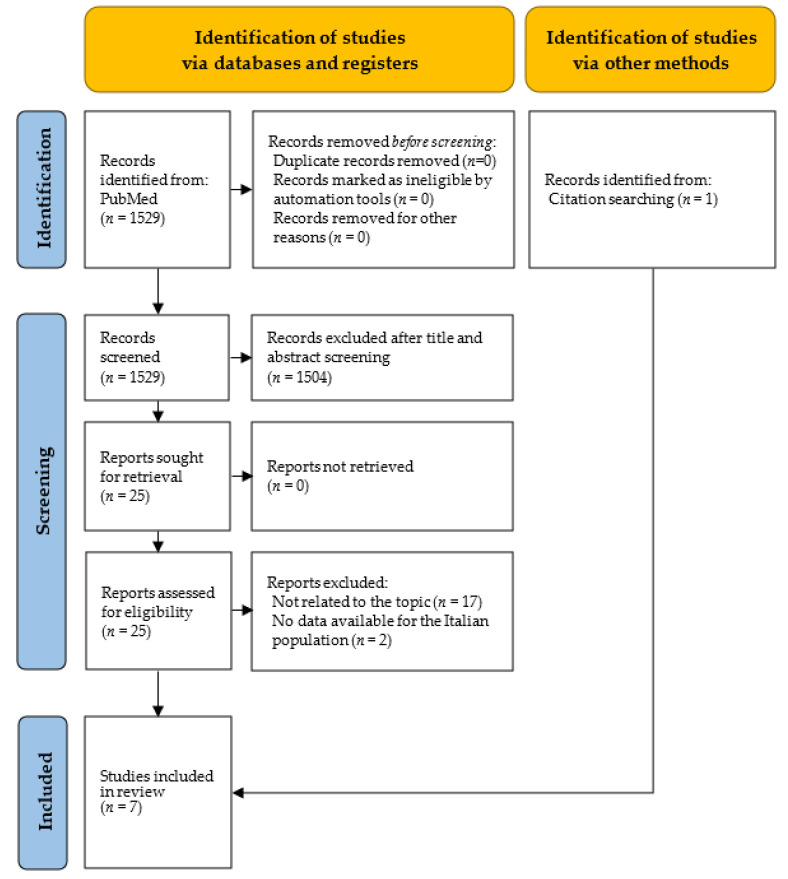
PRISMA flowchart for the systematic review of the impact of vaccinations on COVID-19-related mortality.

**Figure 3 vaccines-11-01011-f003:**
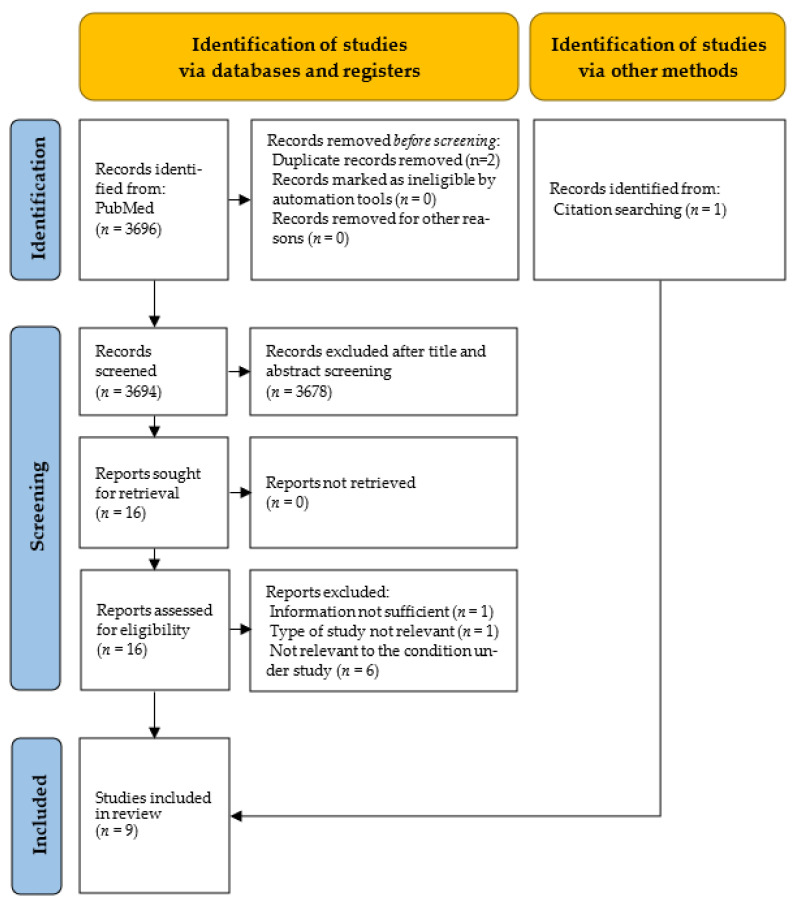
PRISMA flowchart for the systematic review of the impact of vaccinations on COVID-19-related complications.

**Table 1 vaccines-11-01011-t001:** The two search strings.

Topic	Search String	Filters
Effects of vaccination on COVID-19-related mortality	(COVID-19 OR SARS-CoV-2) AND (mortality OR death OR lethal) AND Italy	English language, Adult: 19+ years
Effects of vaccination on COVID-19-related complications	(COVID-19 OR SARS-CoV-2) AND (complication OR sequelae OR hospitalization) AND Italy	English language, Adult: 19+ years

**Table 2 vaccines-11-01011-t002:** Characteristics of the studies investigating the effects of vaccination on COVID-19-related mortality.

Author, Year, [Ref]	Location	Period	Study Design	Sample Characteristics	Study Aim	Vaccine Sources	Results on COVID-19-Related Mortality	Main Findings
Rivasi G, 2021 [42]	Florence, Tuscany region	1 October 2020–31 March 2021, I to III waves	Observational, retrospective	3730 SARS-CoV-2-naïve NHs residents (mean age 84.2, 69% female)	Investigation of impact of vaccination on the course of the pandemic before and after vaccination	Comirnaty	Fully vaccinated subjects showed significantly lower mortality rates than those partially vaccinated and unvaccinated (6%, 18%, and 56%, respectively) *. The death risk was 84% and 96% lower in partially vaccinated subjects (HR 0.157, 95% CI: 0.049–0.491) and in fully vaccinated subjects (HR 0.037, 95% CI: 0.006–0.223) compared to those unvaccinated (HR 0.157, 95% CI: 0.049–0.491) *	Vaccination was followed by a marked decline in mortality among infected residents
Mattiuzzi C, 2022 [43]	Italy	27 December 2020–27 November 2021, II to IV waves	Observational, retrospective	4,487,526 Italian people ≥80 years old	Analysis of the effectiveness of booster doses in older people based on data retrieved from the ongoing nationwide Italian COVID-19 vaccination campaign	Comirnaty, Spikevax, Vaxzervria, Jcovden	An 81% and 98% lower risk ** of death in those who received booster doses compared, respectively, with those who completed the COVID-19 primary vaccination cycle or were unvaccinated	The administration of COVID-19 vaccine booster doses greatly reduced the risk of mortality
Pierobon S, 2022 [44]	Veneto region	I wave 21 February 2020–31 August 2020, II wave 31 August 2020–1 January 2021, III wave 1 January 2021–3 May 2021	Observational, retrospective	852,211 residents in the Veneto region aged between 70 and 100. 33,592 NH residents. After the 1:1 propensity score matching, the size of the two groups was equal to 31 922 cases	Analysis of the risk of hospitalization and death due to COVID-19 among NH residents in comparison with the general older population over time	N/A	During the first wave the RR in NH vs. not NH was 10.10 (95% CI: 8.17–12.47); in the second wave it was of 7.97 (95% CI: 7.17–8.87); in the third wave RR was of 0.48 (95% CI: 0.30–0.77). The overall RR was of 6.07 (95% CI: 5.58–6.61)	By the end of the COVID-19 vaccination program in NHs, the risk of death due to COVID-19 in NH residents was significantly reduced
Russo AG, 2022 [45]	Milano and Lodi, Lombardy region	1 October 2021–31 December 2021, IV wave	Observational, retrospective	2,936,193 ≥19-year-old residents in the provinces of Milan and Lodi	Evaluation of the efficacy of the booster dose in reducing severe SARS-CoV-2 infection in terms of hospital admissions, ICU admissions, and deaths from all causes	N/A	Boosted subjects had an HR of 0.33 (95% CI: 0.29–0.37) for death compared to unvaccinated subjects. HR values in vaccinated subjects for each vaccine dose varied as follows: 1 dose alone, HR 1.80 (95% CI: 1.56–2.09); 1 dose + COVID-19, HR 0.98 (95% CI: 0.82–1.16); 2 doses (<4 months), HR 0.31 (95% CI: 0.26–0.36); 2 doses (4–5 months), HR 0.53 (95% CI: 0.42–0.68); 2 doses (5–6 months), HR 0.57 (95% CI: 0.51–0.64); 2 doses (6–7 months), HR 1.47 (95% CI: 1.31–1.64); 2 doses (7+ months), HR 3.77 (95% CI: 3.52–4.03); 2 doses + COVID-19, HR 1.14 (95% CI: 0.96–1.36)	Two doses received in the previous 5 months and the booster dose significantly reduced the risk of death due to COVID-19
Sacco C, 2021 [46]	Italy	11 January 2021–30 September 2021, III to IV waves	Observational, retrospective	Italian people ≥12 years old	Estimation of the number of averted COVID-19 cases, hospitalizations, and deaths due to vaccination by age group, and geographical macroareas	Comirnaty, Spikevax, Vaxzervria, Jcovden	22,067 averted deaths with vaccination (38% of expected). In particular, in the subjects ≥80 years old (71%), followed by subjects of 70–79 years old (18%), 60–69 years old (8%), and <60 years old (2%)	The largest proportion of deaths prevented by vaccination was observed in the oldest age group (≥80 years). Geographical areas that achieved high vaccination rates faster were able to prevent a larger number of deaths over the summer months.
Pastorino R, 2022 [47]	40 countries	Pre vaccination 21 February 2020–27 December 2020. Vaccination 27 December 2020–30 May 2021 I to IV waves	Observational, retrospective	All countries that had a total of COVID-19 deaths >500 as of end of May 2021	Analysis of the proportion of deaths in different age groups in vaccination versus control periods in different countries	N/A	The COVID-19 deaths in the 0–69 age group compared to the total were of 14% in entire prevaccination period, of 13% in the II wave prevaccination period, and of 16% in vaccination period, respectively	Data show that vaccination was associated with a marked change in the age distribution of COVID-19 deaths in the first 5 months of 2021 in countries that prioritized vaccination among older people, with a relative increase in the share of deaths among nonelderly people
Rivasi G, 2022 [48]	Florence, Tuscany region	1 April 2021–31 October 2021, III to IV waves	Observational, perspective	2271 NH residents (mean age 86.6, 74% female)	Investigation of the long-term impact of vaccination on lethality	Comirnaty	Lethality rate was 5% up to 6 months after the primary vaccination cycle and 8% during the following 3 months	Lethality rate was significant but lower than previously reported in unvaccinated residents of the study area

NH: nursing home; ICU: intensive care unit. * Statistical significance: *p*-value < 0.001. ** Statistical significance: *p*-value < 0.05.

**Table 3 vaccines-11-01011-t003:** Characteristics of the studies investigating the effects of vaccination on COVID-19-related complications.

Author, Year, [Ref.]	Location	Period	Study Design	Sample Characteristics	Study Aim	Vaccine Sources	Results on COVID-19-Related Complications					Main Findings
Rivasi G, 2021 [42]	Florence, Tuscany region	1 October 2020–26 December 2021 (prevaccination period), 27 December 2021–31 March 2021 (postvaccination period), I to III waves	Observational, retrospective	3730 SARS-CoV-2-naïve NHs residents (mean age 84.2; 69% females)	Analysis of the impact of the BNT162b2 mRNA SARS-CoV-2 vaccine on the course of the epidemic in the NHs of the Florence Health District in Tuscany	Comirnaty	COVID-19 Severity * Out of a total of 100 subjects, those unvaccinated had absent or minimal symptoms, mild or severe forms, in 22%, 33% and 44% of cases respectively; the partially vaccinated subjects presented absent or minimal symptoms, mild or severe forms, in 45%, 39% and 16% of cases, respectively; fully vaccinated subjects had no or minimal symptoms, mild or severe forms, in 86%, 14% and 0% of cases, respectively. Hospitalizations ** Out of a total of 100 subjects, 33% of unvaccinated cases, 14% of partially vaccinated cases and 3% of fully vaccinated cases were hospitalized					SARS-CoV-2 vaccination was associated with lower morbidity among infected NH residents
Mattiuzzi C, 2022 [43]	Italy	27 December 2020–27 November 2021, II to IV waves	Observational, retrospective	448,526 subjects ≥80-years old	Analysis of the effectiveness of COVID-19 vaccine booster doses in older people based on data retrieved from the ongoing nationwide Italian COVID-19 vaccination campaign	Comirnaty, Spikevax, Vaxzervria, Jcovden	Hospitalizations: -OR (95% CI) COVID-19-related hospitalizations of unvaccinated compared to Vaccinated (≥5 months), Vaccinated (<5 months), and Vaccinated with booster, was of 0.17 (0.16–0.19; *p* < 0.0019), 0.14 (0.13–0.15; *p* < 0.001), and 0.03 (0.02–0.04; *p* < 0.001), respectively. -OR (95% CI) COVID-19-related hospitalizations of Vaccinated (≥5 months) compared with Vaccinated (<5 months), and Vaccinated with booster, was of 0.80 (0.71–0.90; *p* < 0.001), and of 0.17 (0.14–0.21; *p* < 0.001), respectively. -OR (95% CI) COVID-19-related hospitalizations of Vaccinated (<5 months) compared with Vaccinated with booster, was of 0.21 (0.17–0.26; *p* < 0.001). ICU admissions: -OR (95% CI) COVID-19 related ICU admissions of unvaccinated compared to Vaccinated (≥5 months), Vaccinated (<5 months), and Vaccinated with booster, was of 0.10 (0.07–0.16; *p* < 0.001), 0.14 (0.10–0.21; *p* < 0.001), and 0.02 (0.01–0.04; *p* < 0.001) respectively. -OR (95% CI) COVID-19 related ICU admissions of Vaccinated (≥5 months) compared with Vaccinated (<5 months), and Vaccinated with booster, was of 1.41 (0.82–2.43; *p* = 0.210), and 0.18 (0.06–0.53; *p* < 0.001) respectively. -OR (95% CI) COVID-19 related ICU admissions of Vaccinated (<5 months) compared with Vaccinated with booster, was of 0.13 (0.04–0.37; *p* < 0.001).					The administration of COVID-19 vaccine booster doses is advisable for reducing the risk of morbidity and mortality in older people
Pierobon S, 2022 [44]	Veneto region	I wave 21 February 2020–31 August 2020, II wave 31 August 2020–1 January 2021, III wave 1 January 2021–3 May 2021	Observational, retrospective	852,211 residents in the Veneto region aged between 70 and 100 years (33,592 NH residents and 818,619 non-NH residents).	Analysis of the risk of hospitalization and death due to COVID-19 among NH in comparison with the general older population over time	N/A	RRs of hospitalization for COVID-19 were the following, in I, II, and III wave: −4.89 (95% CI: 4.17–5.74), 2.50 (95% CI: 2.29–2.73) and 0.25 (95% CI: 0.16–0.38), respectively.					The probability of SARS-CoV-2 hospitalization among NH residents had dramatically decreased by the time the COVID-19 immunization campaign in NHs was complete, especially for severe forms of COVID-19
Russo AG, 2022 [45]	Milano and Lodi, Lombardy region	1 October 2021–31 December 2021, IV wave	Observational, retrospective	2,936,193 (≥19 years old) residents in the provinces of Milan and Lodi	Evaluation of the efficacy of the booster dose in reducing severe SARS-CoV-2 infection in terms of hospital admissions, ICU admissions, and deaths from all causes	N/A	HR (95% CI) (reference: unvaccinated) Hospitalizations ICU admissions					Unvaccinated subjects compared to those who received a booster have a 10-fold greater risk of being hospitalized and a 9-fold greater risk of being admitted to ICUs
							1 dose alone	1.34	(1.15–1.56)	0.28	(0.10–0.76)	
							1 dose + COVID-19	0.07	(0.04–0.12)	-		
							2 doses <4 months	0.19	(0.16–0.22)	0.03	(0.01–0.12)	
							2 doses (4–5 months)	0.18	(0.14–0.24)	0.05	(0.01–0.33)	
							2 doses (5–6 months)	0.41	(0.37–0.46)	0.19	(0.12–0.29)	
							2 doses (6–7 months)	0.73	(0.63–0.84)	0.25	(0.13–0.49)	
							2 doses 7+ months	1.65	(1.50–1.82)	0.57	(0.34–0.96)	
							2 doses + COVID-19	0.12	(0.08–0.19)	0.08	(0.01–0.54)	
							Booster dose	0.10	(0.08–0.12)	0.11	(0.03–0.35)	
Sacco C, 2021 [46]	Italy	11 January 2021–30 September 2021, III to IV waves	Observational, retrospective	Italian people ≥12 years old	Estimation of the number of averted COVID-19 cases, hospitalizations, and deaths based on the effect of the vaccination campaign	Comirnaty, Spikevax, Vaxzervria, Jcovden	A total of 79,152 hospitalizations, and 9839 ICU admissions averted by the vaccination campaign. Averted events by age range and percentage (hospitalizations and ICU admissions, respectively) of the total averted events: -subjects <60 years: 23%; 18% -subjects 60–69 years: 17%; 29% -subjects 70–79 years: 19%; 30% -subjects ≥80 years: 41%, 23%					Findings show a positive impact of the COVID-19 vaccination program in Italy and suggest that the rapid vaccination of high-risk groups has prevented a considerable number of severe COVID-19 outcomes
Rivasi G, 2022 [48]	Florence, Tuscany region	1 April 2021–31 October 2021, III to IV waves	Observational, prospective	2271 vaccinated (primary cycle) NH residents (mean age 86.6; 74% females)	Analysis of the long-term impact of BNT162b2 SARS-CoV-2 vaccine on breakthrough infection rates in the NHs of Florence, Italy	Comirnaty	COVID-19 severity Absent/minimal 58% Mild/moderate 25% Severe 17% 8 hospitalizations (8%) (2 on 1 April–31 July and 6 on 1 August–31 October)					Hospitalizations remained stable up to nine months following primary vaccination course
Ripabelli G, 2022 [49]	Molise region	18 October 2021–2 November 2021, IV wave	Observational, prospective	87 subjects (71 NH residents (median age 89; 73% females) and 16 HCWs (median age 52.5; 93.75% females))	Description of the impact of vaccination in terms of hospitalizations during the SARS-CoV-2 outbreak caused by the Delta (B.1.617.2) variant	Comirnaty, Spikevax	38 infected (44%), 3 hospitalizations (3% of the subjects, 8% of the infected, 4% of the residents, 0% of the HCWs), 0 ICU admissions (0%)					A booster dose of mRNA vaccine resulted in high protection against severe disease and hospitalization
Peghin M, 2022 [50]	Udine, Friuli Venezia Giulia region	COVID-19 diagnosis 1 March 2020–31 May 2020 Follow-up at 12 months 1 March 2021–31 May 2021 I wave and III wave	Observational, prospective	At 12 months, 479 patients (53% females), 132 vaccinated with primary cycle (71% females) and 347 unvaccinated (46% females)	Description of the impact of vaccination and the role of humoral responses on post-COVID-19 syndrome 1 year after the onset of SARS-CoV-2	Comirnaty, Spikevax, Vaxzervria, Jcovden	Acute COVID-19 severity * in overall, vaccinated, and unvaccinated subjects, respectively: -Asymptomatic: 8.0%, 14.4%, 5.5%; -Mild: 67.7%, 65.1%, 68.7%; -Severe/critical: 24.3%, 20.5%, 25.8% COVID-19 management in overall, vaccinated, and unvaccinated subjects, respectively: -Outpatient: 71.0%, 75.0%, 69.4%; -Inpatient ward: 24.6%, 22.7%, 25.4%; -Inpatient ICU: 4.4%, 2.3%, 5.2%. Post-COVID-19 syndrome at 6 months in overall, vaccinated, and unvaccinated subjects, respectively: -42.0%, 33.3%, 45.2% Post-COVID-19 symptoms at 12 months compared with post-COVID-19 symptoms at 6 months in Vaccinated and Unvaccinated subjects, respectively: -Unaffected + unchanged: 65.9%, 71.2%; -Worsened: 22.7%, 15.8%; -Improved: 11.4%, 13.0%					Post-COVID-19 syndrome rates are high up to 1 year after acute infection, and receiving the SARS-CoV-2 vaccine is not associated with worsening post-COVID-19 symptoms
Lorenzoni G, 2022 [51]	Veneto region (San Donà di Piave e Jesolo, Ca’ Foncello, Belluno, Dell’Angelo hospital, ULSS 6 Euganea)	May 2021–December 2021, III to IV waves	Observational, retrospective	748 patients (mean age 62 years, 138 (18%) vaccinated (≥2 doses), 58 (8%) partially vaccinated (1 dose), 552 (74%) not vaccinated)	Analysis and comparison of ICU admissions for COVID-19-associated acute respiratory distress syndrome in vaccinated and unvaccinated subjects	N/A	The median time from vaccine administration to ICU admission was **: 22.5 days (IQR, 16.0–49.8 days) for partially vaccinated; 159.0 days (IQR, 112.0–192.0 days) for fully vaccinated. The median time from vaccine administration to hospital admission was **: 16.0 days (IQR, 11.0–32.5 days) for partially vaccinated; 154.0 days (IQR, 110.0–190.0 days) for fully vaccinated.					Vaccinated patients received the second dose of vaccine a median of 5 months before admission to the ICU, whereas for partially vaccinated patients, the median ICU admission time occurred while they awaited the second dose

HCWs: healthcare workers; NH: nursing home; ICU: intensive care unit. * Statistical significance: *p*-value < 0.001. ** Statistical significance: *p*-value < 0.05.

**Table 4 vaccines-11-01011-t004:** COVID symptom severity by vaccination status (percentage of the sample considered by the study).

Article and Study Period	Administered Vaccines	Vaccination Status	Absent/Minimal	Mild	Severe/Critical
Rivasi G, 2022 [48] 1 April 2021–31 October 2021	Comirnaty	FV	58%	25%	17%
Rivasi G, 2021 [42] 1 October 2020–31 March 2021		FV	86%	14%	0%
	Comirnaty	PV	45%	39%	16%
		UV	22%	33%	44%
Peghin M, 2022 [50] 1 March 2020–31 May 2020	Comirnaty, Spikevax, Vaxzervria, Jcovden	FV	14%	65%	21%
		UV	6%	69%	26%

FV—Fully vaccinated (concluded at least primary cycle). PV—Partially vaccinated (at least 1 dose but did not complete the primary vaccination cycle). UV—Unvaccinated.

**Table 5 vaccines-11-01011-t005:** Averted events by age and percentage of the total averted events from 11 January 2021 to 30 September 2021 [46].

Age	COVID-19 Cases	Hospitalizations	ICU Admissions
<60 ^1^	62.7%	23.3%	18.1%
60–69	12.0%	16.6%	28.6%
70–79	9.2%	19.2%	30.2%
≥80	15.9%	40.7%	22.8%

^1^ Including subjects from 12 years old.

## Data Availability

The data are contained within the article.

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
