# Peer review of "The Impact of Vaccination on COVID-19 Burden of Disease in the Adult and Elderly Population: A Systematic Review of Italian Evidence"

_vaccines, 2023, doi:10.3390/vaccines11051011_

Round 1
Reviewer 1 Report
The systematic review investigates the effect of vaccination against SARS-CoV-2 infection on COVID-19 disease, complications, and deaths in Italian population. The paragraphs of limitations of the study, correctly included in the discussion section, help the reader to understood better the capacity of authors to put together different works, but the objective limit to obtain a generalized conclusion due the use of different vaccines/protocol/population/timing/circulating virus variants.
Some points need to be better clarified in the text:
- - Use SARS-CoV-2 instead of COVID-19 when it refers to “infection”. COVID-19 refers to the name of the disease.
- - Define fully and partially vaccination protocol, variable among different countries.
- - Specify how all vaccines adopted in different studies have “a good protective effectiveness”: are they capable to elicit similar level of antibodies or memory B cells? This point can be supported by references to evidence the different critical role of different immune arms.
- - The post COVID syndrome paragraph includes only one published work. This is a strong limit in a systematic review that need to be addressed in the text.
minor revisions required
Author Response
Dear Editor and Dear Reviewer,
We would like to thank you for the opportunity to resubmit our work. We have amended the paper according to the received suggestions and we hope that it now appears improved.
In the attachment, point-to-point answers are provided.
Best regards,
The Authors

Reviewer 2 Report
The authors conducted two systematic reviews of non-randomized studies of the effects of Covid-19 vaccination on Covid-19 related complications (9 articles) and deaths (7 articles) in the Italian population based on data gathered from before the start of the vaccination campaign in Italy (12/27/2020) until early 2022 in accordance with the PRISMA guidelines.
Overall, this review is fairly well written and organized. The studies are summarized in several detailed tables and discussed in the text. Nevertheless, the authors did not clearly correlate the timeline of Italy’s immunization plan, the availability of vaccines, hospital beds and ventilators for different groups in different regions, and the predominant Covid-19 variants present at the time with the non-randomized study findings. Such correlations would allow the readers to better compare the findings of the studies in their context. For example, Figure 1 could be revised to include the above information for each systematic study separately.
Line 47-52: present the Covid-19 complications most frequently encountered Italy, starting with the most common ones.
Line 67: which were the five Covid-19 vaccine candidates, and which ones were administered to the Italian population?
Line 78-80: provide a detailed timeline of the immunization plan through early 2022 before jumping to the final numbers of vaccinated individuals.
Line 115-121: for each criterion, indicate how many studies were excluded or retained.
Line 163, 221, etc.: what were the most common indications for hospitalizations and ICU admissions?
References: update the list to include articles from 2023, as appropriate.
Figure 1: which of these ten studies have to do with Covid-19 related complications (9 articles) and deaths (7 articles)? Clarify the dates: 03/2, 05/2, 07/2 on the x axis? Please use a standard date format.
Line 172: this should be Figure 2. Detail the reason(s) why 1504 records were excluded. Similarly in Figure 4, detail the reason(s) why 3678 records were excluded.
Table 2: indicate the time period of each study. Also, indicate the source of the vaccine (Pfizer/BioNTech, Moderna, others; mix of different vaccines).
Table 3: indicate the time period of the study.
Table 4: indicate to which Covid-19 wave(s) each period correspond to. Regarding the sample characteristics, indicate the mean or median age and range (e.g., not “>=80”). Indicate the source of the vaccine. Indicate which differences were statistically significant.
Table 5: see comments for Table 4. Also, indicate the type of hospitalization (isolation room, ICU…) and the medical reason(s) for the hospitalization.
Overall, the quality of the English language is very good. Occasionally, some sentences should be reworded to read as written by a native speaker.
Author Response

(The authors gave the same response as above.)

Reviewer 3 Report
The paper describes a systematic review of the impact of SARS-CoV-2 vaccination on mortality, hospitalization, disease severity, and post-COVID syndrome in Italy. The following was noted:
I would recommend that the authors look at the latest paper from Page et al., 2021 “The PRISMA 2020 statement: an updated guideline for reporting systematic reviews” again and ensure that the submitted paper checks all criteria for the 2020 statement. Missing data in the abstract (3, 5, 6, 7, 11,12) and other information (24a-27) was noted and the paper would benefit from a revision based more closely on the 2020 guidelines.
Lines 147-169 compare the various studies included in the systematic review but do not describe the effects of vaccination on COVID-19 mortality. Perhaps place this paragraph under a different heading i.e. comparison of studies included in the systematic review on COVID-19 mortality. Please refer to Table 4 in this section as it contains the data abstracted from the papers and defines the variables collected.
Figure 3 is from the paper by Pierobon et al. The data has been reported in the text and could be deleted without affecting readability or analysis.
Lines 248-251 – sentence meaning is unclear, please rewrite.
Please ensure consistency in number of places reported after the decimal. In the first section, only whole numbers are reported e.g. 60% while in later sections one decimal is reported e.g. 60.4%. Please pick one decimal format or percentages and maintain throughout.
Lines 254-279 - compare the various studies included in the systematic review but do not describe the effects of vaccination on COVID-19 related complications. Perhaps place this paragraph under a different heading i.e. comparison of studies included in the systematic review on COVID-19 related complications. Please refer to Table 5 in this section as it contains the data abstracted from the papers and defines the variables collected
There is repetition in sections on COVID-19 deaths and COVID-19 hospitalizations. E.g. lines 189-192 and lines 347-350; lines 214-217 and lines 354 – 357. Please revise.
Table 5 should be revised for spacing and readability.

The paper would benefit from some editing for English as some of the words used were not appropriate and some of the sentence construction clumsy or overly long. Some language editing has been performed on the electronic version of the paper but additional editing for language may be required.
Author Response

(The authors gave the same response as above.)

Round 2
Reviewer 2 Report
Figure 1. thank you for indicating which studies are related to Covid-19 complications and death. However, the superscript you use is really minuscule - please use a more conspicuous way to identify the studies.
Table 2. some of the text in the two right-most columns overlap and is incomplete.
Recommend minor editing of English language.
Author Response
Dear Reviewer,
we have amended the paper according to the received suggestions and we hope that it now appears improved.
In the attachment, point-to-point answers are provided.
Best regards,
The Authors

Reviewer 3 Report
Line 62: "Published data suggests..."
Line 82: "Since then, the country has a approved the use of other vaccine."
Line 105: "...such as influenza vaccines..."
Line 113-114: "Ongoing monitoring of the effectiveness and safety of COVID-19 vaccination is necessary as it may be influenced by features like waning immunity and the emergence of more transmissible variants."
Line 276: "Sacco et al. [46] conducted a study to evaluate the impact of COVID-19 vaccination on the prevention of infections, hospitalizations, ICU admissions, and deaths."
Quality of English language has been improved.
Author Response

(The authors gave the same response as above.)
